# High Throughput strategies Aimed at Closing the GAP in Our Knowledge of Rho GTPase Signaling

**DOI:** 10.3390/cells9061430

**Published:** 2020-06-09

**Authors:** Manel Dahmene, Laura Quirion, Mélanie Laurin

**Affiliations:** 1Oncology Division, CHU de Québec–Université Laval Research Center, Québec, QC G1V 4G2, Canada; manel.dahmene@crchudequebec.ulaval.ca; 2Montréal Clinical Research Institute (IRCM), Montréal, QC H2W 1R7, Canada; Laura.Quirion@ircm.qc.ca; 3Université Laval Cancer Research Center, Québec, QC G1R 3S3, Canada

**Keywords:** Rho GTPase 1, RhoGEF 2, RhoGAP 3, RhoGDI 4, high throughput screening 5, proximity labelling 6

## Abstract

Since their discovery, Rho GTPases have emerged as key regulators of cytoskeletal dynamics. In humans, there are 20 Rho GTPases and more than 150 regulators that belong to the RhoGEF, RhoGAP, and RhoGDI families. Throughout development, Rho GTPases choregraph a plethora of cellular processes essential for cellular migration, cell–cell junctions, and cell polarity assembly. Rho GTPases are also significant mediators of cancer cell invasion. Nevertheless, to date only a few molecules from these intricate signaling networks have been studied in depth, which has prevented appreciation for the full scope of Rho GTPases’ biological functions. Given the large complexity involved, system level studies are required to fully grasp the extent of their biological roles and regulation. Recently, several groups have tackled this challenge by using proteomic approaches to map the full repertoire of Rho GTPases and Rho regulators protein interactions. These studies have provided in-depth understanding of Rho regulators specificity and have contributed to expand Rho GTPases’ effector portfolio. Additionally, new roles for understudied family members were unraveled using high throughput screening strategies using cell culture models and mouse embryos. In this review, we highlight theses latest large-scale efforts, and we discuss the emerging opportunities that may lead to the next wave of discoveries.

## 1. Introduction

Rho GTPases, which are best known for their regulation of the actin cytoskeleton, play central roles in many physiological and pathological processes [1]. Due to their capacity to orchestrate the formation of tissue architecture, Rho GTPases have emerged as fundamental regulators of morphogenesis during embryonic development [2]. Their ability to coordinate cellular motility also makes them key mediators of tumor invasion. Moreover, since recent deep sequencing efforts have identified recurrent mutations in Rho GTPases in cancer, these molecules are rising as attractive therapeutic targets for the design of new cancer treatment regimens [3].

Rho GTPases are part of the Ras superfamily of small GTPases [4,5,6]. In humans, there are 20 Rho GTPases. Until now, a majority of their study has focused on signaling by the three prototypical members, RAC1, RHOA, and CDC42, which have long been appreciated for their respective contributions to lamellipodia, stress fiber, and filipodia formation [7]. While it is clear that the roles of Rho GTPase proteins extend far beyond cytoskeletal regulation, we still do not fully understand the complete spectrum of their biological functions. In fact, most of our understanding of Rho GTPases’ roles and regulation mechanisms has emerged from in vitro studies, during which single gene knockdown or overexpression of mutant forms were achieved in cell lines. As such, several family members remain understudied. Moreover, we still do not fully understand how different stimuli that converge on the same Rho GTPase trigger different responses in cells and the repertoire of Rho GTPases effectors remains incomplete. Given the complexity involved and the important crosstalk between Rho GTPases signaling networks, only large-scale global approaches are likely to mitigate for this current gap in knowledge. Recently, unbiased proteomic methodologies have contributed to significantly expand our understanding of Rho GTPases signaling interactions. Additionally, several groups have exploited high throughput screening approaches to identify new functional roles for some of the most understudied Rho GTPases network components. Here, we discuss the new insights provided by these system level approaches, and we highlight some of the remaining challenges the Rho GTPases field is facing.

## 2. Rho GTPases Cycle and Their Regulation

In mammals, there are 20 Rho GTPases divided into eight subfamilies (Figure 1). Classical Rho GTPases from the CDC42, RAC, RHO, and RHOF subfamilies act as molecular switches that cycle between an inactive GDP-bound and an active GTP-bound conformation. These Rho GTPases are largely regulated by the exchange of their GDP/GTP bound state. The four remaining subfamilies, namely the RHOU/RHOV, RND, RHOH, and RHOBTB subfamilies (Figure 1), are considered atypical because they either do not follow this canonical cycle of regulation or because they have additional structural and functional features that distinguish them from the classical Rho GTPases [8]. More specifically, RHOU and RHOV have unusual GDP/GTP cycling rate, and they are believed to be predominantly bound to GTP in cells due to their high intrinsic guanine nucleotide exchange activity when compared to CDC42 [9,10,11]. Intriguingly, RHOU has been shown to be highly regulated at the transcriptional level through the activation of several developmental pathways, such as WNT and NOTCH [11,12,13,14,15,16]. The RHOH and RND subfamilies do not cycle between GDP and GTP due to their lack of intrinsic GTPase activity and are therefore constitutively bound to GTP and active in cells [17,18]. The current view is that RHOH acts in cells to antagonize the action of the classical Rho GTPases [19,20,21] while RND proteins have been shown to have antagonistic effects with RHOA [17,22]. Lastly, RHOBTB GTPases are much larger than other Rho GTPases due to the presence of several additional domains, and their activation is thought to be partly regulated via protein–protein interactions that release their auto-inhibited conformation [23,24,25,26].

In addition to the 20 Rho GTPases found in human, RHOT1 and RHOT2, also known as MIRO-1 and 2 (for mitochondrial Rho), were identified during a search for proteins that contain a Rho consensus domain [27]. Due to this feature, these GTPases were initially classified as part of the atypical Rho GTPases subgroup [27,28]. RHOT1 and RHOT2 localize at the mitochondria where they regulate several aspects of mitochondria homeostasis and transport [28]. Due to their large sequence divergence, subsequent studies have since reclassified these two GTPases as a distinct subgroup in the Ras superfamily along with the Ras, Rho, Ran, Rab, and Arf GTPases [5].

Classical Rho GTPases are regulated by three families of proteins that entail the guanine nucleotide exchange factors (RhoGEFs), the GTPase-activating proteins (RhoGAPs), and the guanine nucleotide dissociation inhibitors (RhoGDIs) [29]. By binding to Rho GTPases, RhoGEFs favor GDP and Mg^2+^ dissociation. RhoGEFs then stabilize the Rho GTPases in nucleotide-free form that rapidly associate with GTP and dissociate from the RhoGEFs as a consequence of the relatively high concentration of GTP in cells when compared to GDP concentration (Figure 2) [30,31,32,33,34]. RhoGEFs are divided into two subfamilies, the Dbl-like and the DOCK subfamily, according to their respective GEF catalytic domain [30,35]. Upon GTP binding, Rho GTPases undergo a conformation change that enables their interaction with specific effectors, which promotes downstream signaling and ultimately triggers a biological output in cells (Figure 2) [36]. By interacting with a definite set of effectors and by bringing these effectors in the vicinity of the Rho GTPases they are activating, RhoGEFs play an important scaffolding function and orchestrate downstream signaling in response to an upstream cell stimulus (Figure 2) [35]. In contrast, RhoGAPs bind to active Rho GTPases and stimulate the Rho GTPases’ weak intrinsic GTPase activity. This promotes GTP hydrolysis and inactivate the Rho GTPases (Figure 2) [37]. Throughout this cycle, the majority of Rho GTPases are modified via the addition of isoprenoid lipids at the first cysteine residue of their C-terminal *CAAX* motif [38]. This posttranslational modification facilitates their association with membranes. Still, this lipid modification is not sufficient to determine the subcellular localization of Rho GTPases. This requires a second C-terminal signal in the Rho GTPases hypervariable region that allows targeting to specific membrane compartments [39,40]. Over the years, RhoGDIs have been generally recognized as Rho GTPases’ negative regulators. Notably, RhoGDIs allow the removal of Rho GTPases from membranes by binding to the GDP-bound form, which in a second step promote the transfer of the Rho GTPase’ C-terminal lipid extension from the lipid bilayer into the RhoGDI hydrophobic pocket [41,42,43,44,45,46,47]. This allows RhoGDIs to sequester Rho GTPases in the cytoplasm and to prevent their premature reactivation [48]. Yet, RhoGDIs have also been shown to stabilize the Rho GTPases proteins in the cytoplasm and to prevent their degradation via the proteasome [49]. While several studies have showed that RhoGDIs can bind active and inactive Rho GTPases in vitro [50,51,52,53,54,55,56], these regulators were found to be mainly in complex with the inactive GDP-bound Rho GTPases in cells [57]. By using membrane extraction assays and by visualizing Rho GTPases activity in living cells, Golding et al. showed that RhoGDIs are able to extract active GTP-bound Rho GTPases from membranes (Figure 2), a process proposed to be an important mechanism to allow spatiotemporal concentration of Rho GTPases in cells [58,59]. Finally, Rho GTPases can be regulated by their gene expression and they are also extensively modified via posttranslational modifications, which include phosphorylation, lipid addition, ubiquitination, and SUMOylation [60]. These modifications alter Rho GTPases localization, activity, and stability and, ultimately, fine-tune their signaling responses to a specific stimulus (Figure 2).

While Rho GTPases themselves are relatively simple molecules, the intricacy of Rho GTPases signaling networks is attributed to the large number of regulators and effectors that by far exceeds the number of Rho GTPases themselves. In mammals, we count 80 RhoGEFs, 69 RhoGAPs, 3 RhoGDIs, and a substantial number of effector proteins [29]. The complexity of these networks is further increased by the important crosstalk between them. Therefore, only large scale unbiased global approaches can help unravel all the signaling activities played by these networks.

## 3. Complex Rho GTPases signaling Hubs Are Revealed by proteomic Approaches

Rho GTPases are required to bind to their protein regulators and effectors to integrate upstream stimuli and relay downstream signals. Yet, the identification of Rho GTPases protein partners has always been challenging due to their strong association with membranes and cytoskeletal components. The emergence of large-scale proteomic approaches has paved the way toward a more global view of these wide protein networks (Table 1). In a first study, Paul et al. developed a quantitative GTPase affinity purification (qGAP) combined with mass spectrometry strategy to identify protein partners of CDC42, RAC1, RHOA, RHOB, RHOC, and RHOD [61]. Briefly, beads loaded with each recombinant Rho GTPases were loaded with GDP or GTPγ and used to pull-down proteins from differentially SILAC (stable isotope labeling by amino acids in cell culture)-labeled HeLa cell lysates. More than 1000 proteins mainly enriched in Rho GTPases effectors were identified using this strategy. While this approach was compatible with the use of animal tissue, the uncovered interactions occurred in the context of cellular lysates. Whether these remained relevant in living cells still had to be determined.

The more recent unbiased proximity biotinylation (BioID) mass spectrometry-based strategies allow investigators to identify protein networks occurring directly in living cells. The BioID method relies on the fusion of a promiscuous version of *Escherichia coli* biotin ligase (BirA*) to a protein of interest [62,63]. Upon expression of the fusion protein in cells in the presence of biotin, endogenous proteins that are proximal to the bait are biotinylated on available lysine residues and recovered through streptavidin pull-down. Importantly, this allows for the identification of transient interactions as well as those that can take place in insoluble cellular compartments. A BioID approach was used to systematically define the proximity interactome of Rho GTPases in their wild-type, nucleotide-free, and active forms to obtain a global view of these wide networks [64]. Impressively, close to 10,000 proximal interactions with Rho GTPases in two cell lines (HEK293 and HeLa cells) were revealed. By using the nucleotide-free form of RHOA, RHOG, RAC1, and CDC42, that can trap RhoGEFs in cells, and the active form of all Rho GTPases that interact with RhoGAPs, the fundamental question of Rho regulators specificity was addressed [64]. With the help of this clever trick, it was revealed that a large fraction of RhoGEFs are highly specific and that their activity is limited toward a few Rho GTPases (Figure 3a). This is the case notably for the RHOA-specific RhoGEFs, ARHGEF1, ARHGEF2, ARHGEF5, ARHGEF17, NGEF, ECT2, and AKAP13 (Figure 3b). Still, some RhoGEFs such as FARP1 can bind to the nucleotide-free version of the three prototypical Rho GTPases, RAC1, RHO, and CDC42 (Figure 3b). As for RhoGAPs, the use of Rho GTPases active form (always bound to GTP), confirmed their previously described promiscuity (Figure 3c) [65]. RhoGAPs have the ability to bind and inactivate a broad spectrum of Rho GTPases from the same or from different subfamilies (Figure 3d) [64]. Together, this reinforce the model in which RhoGEFs, by interacting simultaneously with a limited set of Rho GTPases and a specific set of effectors, become key players to relay a precise downstream cellular response in response to a stimulation.

Over the years, several studies using various fluorescent biosensors have beautifully highlighted the spatiotemporal regulation of Rho GTPases in cells [66,67,68,69,70,71]. Notably, researchers were able to visualize concentric zones of CDC42 and RHOA activation during wound healing of *Xenopus* oocytes [67]. Distinct patterns of RAC1/2 and CDC42 activation were also observed during phagocytosis [72]. Still, understanding how these highly specific patterns are generated remains a challenge. Rho GTPases regulators are likely to play an important role in this process. The visualization of the steady-state distribution of RhoGEFs and RhoGAPs in MDCK epithelial cells by confocal live-cell imaging revealed the presence of these proteins in virtually all cellular compartments, which would allow them to regulate Rho GTPases in all of these locations [73]. Among the 77 RhoGEFs and 66 RhoGAPs tested, a majority of these proteins was found enriched in specific cellular structures rather than broadly distributed in cells [73]. Intriguingly, structures that were not known to harbor Rho GTPases signaling, such as endomembrane, the Golgi, and mitochondria, only hosted RhoGAPs (Figure 4). One hypothesis, is that the RhoGAPs located in these locations display a housekeeping function, and they allow the inactivation of GTP-bound Rho GTPases that would have diffused away from their desired site of action [73].

During cell spreading, active and inactive zones of RAC have been described (Figure 4) [74]. At the core of this process, focal adhesions are multi-protein structures that link the actin cytoskeleton with the extracellular matrix. Rho GTPases regulation is essential to focal adhesion dynamics, and quite strikingly, 25% of Rho regulators can be found at these adhesion sites [73]. Importantly, by measuring RhoGEFs and RhoGAPs localization, RAC1-GEFs were shown to localize preferentially at nascent adhesions along the cell periphery, while RAC1-GAPs were shown to be located along more mature focal adhesions in the center of cells (Figure 4). This segregation of regulators contributes to establish active and inactive zones of Rho GTPase signaling [73]. Overall, these results support the idea that a large fraction of Rho GTPases’ spatiotemporal regulation is conveyed by the regulators themselves that delimit the diffusion of the Rho GTPases forms in cells (Figure 4).

Establishing the interactome of RhoGEFs and RhoGAPs has also revealed the existence of important homotypic interactions between these regulatory proteins [73]. Several interactions were confirmed between RhoGEFs and RhoGAPs, in between RhoGEFs, while fewer interactions were detected in between the RhoGAPs themselves. The association of regulatory proteins in complexes increases their combinatorial possibilities to control downstream signaling and enables a coordination of cellular responses through the crosstalk between Rho GTPases. This important interplay emphasizes the value of studying the Rho GTPases networks from a larger perspective. Altogether, all of these publicly available interactome data sets are a treasure trove of information likely to be highly valuable resources for the Rho GTPases community [64,73].

## 4. High throughput screening Strategies Uncover New Functions for Rho GTPases Network Components

Over the years, several screens have contributed to identify new genes involved in cell migration, and it is not surprising that several Rho GTPases were amongst the unearthed molecules [75,76,77,78,79]. More recently, several groups have relied on RNA interference (RNAi) screens to systematically test the requirement of Rho GTPases, RhoGEFs, RhoGAPs, and RhoGDIs in distinct biological processes. These approaches facilitated the discovery of new roles for understudied Rho GTPases family members (Table 1).

Rho GTPases signaling networks have emerged as key regulators of tumor invasion, and the use of unbiased high throughput screening approaches has largely contributed to delineate the role of individual network components during specific steps of the invasion process. Two studies by the group of Christopher Marshall pioneered this approach and contributed to highlighting that cancer cells rely on distinctive regulators to mediate different migration modes [80,81]. The crucial requirement of RhoGEFs during cellular invasion was emphasized when each of these Rho GTPases regulators was tested for its ability to regulate the amoeboid or the mesenchymal motility of melanoma cells [80,81]. Amoeboid movement is characterized by the blebbing of cells through the extracellular matrix without proteolysis. This migration mode requires high levels of actomyosin contractility, and it differs from mesenchymal movement, which is characterized by cells with an elongated morphology that assemble a leading edge [82]. Cancer cells are known to alternate between these modes as a way to adapt to their microenvironment [83]. In these screens, several RhoGEFs were shown to control melanoma cell migration. Still, only DOCK10 regulated ameboid features. In fact, silencing of DOCK10 led cells to switch migration mode and to instead rely on mesenchymal motility to invade a 3D environment [80]. In contrast, DOCK3 was shown to rather regulate cells amoeboid features and silencing of DOCK3 increased the proportion of melanoma cells that invade a 3D matrices in an amoeboid manner [81]. The ability of cancer cells to alternate modes of motility driven by different Rho GTPases as a way to adapt to their surrounding raises the important issue as to why MMP protease treatments may not be therapeutically viable in targeting cancer metastasis [84,85]. Altogether, these studies underscore the many challenges associated with the treatment of metastatic cancer patients.

Since, several groups have harnessed similar approaches. With the aim of isolating regulators of prostate cancer cell migration, Tajadura-Ortega et al., designed an RNAi screen targeting 202 genes among the Rho GTPases network components [21]. By using wound healing assays and threshold image analyses, they revealed that 25% of Rho GTPases signaling molecules contribute either positively or negatively to the migration of prostate cancer cells [21]. The large fraction of regulators revealed by this screen emphasizes the low level of functional redundancy amongst this family as well as the key role played by these factors. One unexpected hit from this screen was RHOH, whose role had been thought thus far to be restricted to hematopoietic cells [19,20]. Depletion of RHOH was shown to reduce the speed and the persistence of prostate cancer cells. Intriguingly, RHOH expression is not limited to prostate cancer cells, but this Rho GTPase is broadly expressed in other epithelial cancer cell lines, which suggest it might also contributes to cancer progression in these contexts [21]. In another study, Pascual-Vargas et al., rather focused on identifying RhoGEFs and RhoGAPs that orchestrate the morphology of highly metastatic breast cancer cells. For this, 142 genes among the Rho GTPases regulators were depleted using RNAi in one poorly and one highly metastatic breast cancer cell line, i.e., the MDA-MB231 and the LM2 cells, respectively. Following depletion, a total of 127 individual features characterizing the cellular shape and the activation of the YAP signaling pathway were measured for each condition. This large dataset was further probed and a list of top hits for each parameter was identified for more in-depth analyses [86]. Kang et al. rather focused on identifying RhoGAPs that contribute to epithelial to mesenchymal transition (EMT), a process often correlated with metastasis progression [87]. To do so, they used the MCF10A cell lines and revealed that 57 RhoGAPs were expressed in these cells. Using siRNA, they individually silenced the 57 RhoGAPs expressed and characterized in detail the MCF10A cell morphology. Using this approach, they revealed that depletion of 15 RhoGAPs led cells to adopt a spindle-like morphology in comparison to their baseline polygonal shape. Altogether, these screens underline that signaling specificity is often controlled by the Rho GTPases regulators themselves.

Screening approaches in mammalian cells were also useful to probe the function of Rho GTPases in physiological processes. Notably, Rho GTPases network components involved in thrombin-induced endothelial permeability in primary umbilical vein endothelial cells were identified using a siRNA screen that functionally tested 270 human Rho GTPases and Rho-associated genes [88]. A total of 15 top hits that modify the response to thrombin were identified [88]. Intriguingly, the depletion of the Rho GTPase RHOD led to the most potent disruption of endothelial barrier integrity, which revealed its important role in this process [88]. In another study, Zaritsky et al. rather designed a shRNA screen targeting 80 RhoGEFs in human bronchial epithelial cell monolayers to identify the ones that regulate the long-range collective migration in these cells [89]. Quite strikingly, 75 RhoGEFs were shown to be expressed in bronchial epithelial cells, which allows tremendous functional specification. A total of 10 RhoGEFs were revealed as specific regulators of the collective migration of these cells. In general, these in vitro screening approaches offered a powerful lens through which to examine the cellular response to specific stimuli and when looking at a predetermined cellular phenotype. While all of these approaches certainly revealed important functions of Rho GTPases network components, they all still rely on in vitro culture models and further validations will be required to see if these findings translate in vivo.

The development of in vivo models is in fact crucial since not all biological processes, such as the complex cellular rearrangements required for morphogenesis, can be recapitulated using in vitro cell culture models. Due to their ability to coordinate cytoskeletal dynamics, Rho GTPases have emerged as key regulators of morphogenesis [2]. Yet, only a fraction of Rho GTPases network components has been studied in regard to their ability to orchestrate embryonic development in mammals [90]. It is therefore important to use in vivo models to test the requirement of all Rho GTPases network components during morphogenesis. The mouse skin is an excellent system to tackle this challenge since the skin rely on complex cytoskeletal rearrangements for its formation [91,92,93]. Additionally, the development of an ultrasound-guided method of in utero lentiviral injection that allows the specific transduction of mouse skin progenitors has opened the door for high throughput studies that are otherwise impossible using conventional knockout strategies [94,95,96,97,98,99,100,101]. Intriguingly, majority of the Rho GTPases, RhoGEFs, RhoGAPs, and RhoGDIs are expressed in mouse skin progenitors during embryonic development suggesting the important contributions of these molecules to skin formation [97,102,103]. By building a lentiviral shRNA library targeting 166 genes from Rho GTPases network components, namely 20 Rho GTPases, 77 RhoGEFs, 66 RhoGAPs, and 3 RhoGDIs; and by taking advantage of the lentivirus delivery system that targets skin progenitors [94], we designed and validated a novel skin morphogenesis screen in live mouse embryos [97]. Our screen revealed that 42% of the genes targeted are essential for proper skin development. More specifically, seven genes were candidate regulators of epidermal differentiation, 26 genes acted as regulators of hair follicle development and 35 genes were candidate regulators of both processes. The validation of several candidates revealed that these were essential for a plethora of steps required for proper skin development. Notably, we revealed that the RhoGEF FGD2 is required for hair follicle specification and that hair follicle downgrowth is perturbed in the absence of the RhoGEF TRIO. Altogether, the variety and the large number of regulators uncovered using an in vivo RNAi morphogenesis screen emphasizes the power of this strategy as well as the low level of functional redundancy between Rho GTPases network components.

Among the top hits identified in the in vivo morphogenesis screen, the atypical Rho GTPases RHOU was revealed as a key regulator of the establishment of planar cell polarity (PCP) cues in the skin [97]. PCP signaling pathways have been shown to orchestrate hair follicle orientation in the developing skin [104] and defects in PCP organization and hair follicle orientation are often associated with perturbation of cellular contractility in epidermal cells [104,105,106,107,108]. In order to understand the molecular mechanism by which RHOU acts in the skin, the proximity interactome of this atypical Rho GTPase in primary mouse keratinocytes was determined. PAK2 was shown to be the most abundant partner in these cells. Overexpression of RHOU in keratinocytes triggers PAK1/2 activation, which inversely correlated with the phosphorylation of myosin light chain 2. Altogether, the proteomic analyses revealed that RHOU interacts with proteins that shuttle between focal adhesions and cell–cell junctions. By regulating the assembly of these structures, RHOU is required to generate a prototypical epidermal shape in skin progenitors that is required for PCP establishment. Additionally, RHOU acts to restrict hair follicle downgrowth by maintaining epidermal integrity; a similar mechanism is also utilized in the foregut endoderm to restrict the formation of organ primordia [14]. Our findings underscore the advantages of combining large-scale screening and proteomic studies to simultaneously gain insights into the functional and molecular mechanism of top screen hits.

## 5. Discussion

Recent interactome approaches have provided us with a clearer picture of the intricate signaling hubs that assemble around Rho GTPases. Still, our understanding of these modules’ dynamics is far from being reached. In particular, it remains to be determined how the Rho GTPases interactome is reshuffled when cells integrate various stimuli and when they are surrounded by different micro-environments. Notably, Rho GTPases have been shown to be highly post-translationally modified via ubiquitination and phosphorylation [60]. These modifications are likely potent ways to dynamically remodel their interactome. Moreover, the extent of the crosstalk between Rho GTPases signaling hubs is not fully understood. Only global network analyses following cellular treatments will resolve these issues. These questions will also certainly benefit from new enzyme-catalyzed proximity labeling methods such as miniTurbo, TurboID, and APEX, which allow the identification of quick temporal changes in protein networks subsequent to cellular stimulations [109,110,111]. Inversely, these approaches could be taken advantage of to elucidate the signals required to activate specific pathways. The recent interactome studies have also proved useful in effector landscape definition, and they have contributed to significatively increasing the size of Rho GTPases’ effector repertoire. One important aspect to keep in mind is that all of the proximity labelling approaches allow the identification of proteins that are present in a define perimeter of the protein bait in cells. Therefore, significant non-direct interactions are also revealed by these strategies. Still, by digging into the uncharted datasets, the list of cellular functions regulated by Rho GTPases networks is likely to expand greatly in the near future.

A critical aspect exposed by interactome approaches is the necessity of generating physiological models to study Rho GTPases’ functions. Remarkably, Bagci et al. revealed that only 33% of the Rho GTPases proximity interactions were shared between HEK293 and HeLa cells lines, which highlights that cellular context matters deeply [64]. Since not all biological processes can be recapitulated in vitro, we might still be severely underestimating the biological functions played by these proteins. Therefore, it is instrumental to develop high throughput methods that are compatible with physiological and disease in vivo models. Notably, organoid models have the potential to form a bridge between in vitro and in vivo studies and would allow to get a handle on physiological relevance. Still, even if siRNA and shRNA screens were to move to organoids or in vivo models, these types of screens would continue to have limitations. Indeed, while false positive hits can be reduced in these screens by using multiple siRNAs or shRNAs that target the same gene, false negative hits attributed to a failure to properly knockdown a target of interest will remain an issue. Therefore, full knockout approaches will always have their relevance in follow up studies to investigate biological functions of genes.

Finally, it is becoming clear that Rho GTPases signaling plays a crucial role during tumor progression and that signaling by Rho GTPases is perturbed by a wide range of mechanisms in cancer [3]. One indirect way by which Rho GTPases signaling is disrupted is through their regulators, the RhoGEFs and RhoGAPs [3,112]. Mutations of these regulators as well as changes in their expression levels have been shown to either promote or suppress tumor progression and growth [3,112,113]. Overall, these perturbations are likely to affect Rho GTPases’ interactome. The expression of the Rho GTPases RAC1, RHOA, and CDC42 is also altered in several cancer types [75,114,115,116,117,118,119,120,121,122,123,124,125,126]. The recent efforts in next-generation sequencing have revealed that Rho GTPases are mutated in a variety of cancers, and recurrent mutations in RAC1 and RHOA have been identified [3,127,128,129,130,131,132,133,134,135]. Intriguingly, several mutations identified in RAC1, namely RAC1^P29S^, RAC1^N92I^, and RAC1^C157Y^, create a fast cycling Rho GTPase with an enhanced exchange capacity for GDP and GTP [128,129]. While these mutations are thought to enhance RAC1 interaction with its effectors, it would be worth investigating if some de novo interactions are created and if some are lost. Similarly, the various mutations identified in RHOA in cancer cells are likely to perturb its interactome [136]. Altogether, gaining a broader picture of these changes could clarify how these mutated Rho GTPases contribute to tumor progression, and ultimately, this could lead to new therapeutic opportunities.

## Figures and Tables

**Figure 1 cells-09-01430-f001:**
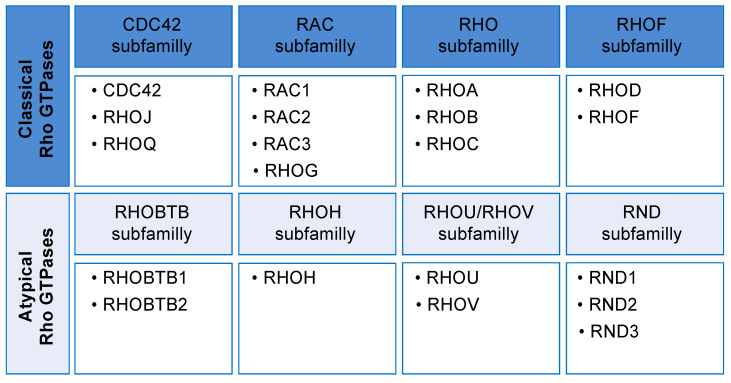
Rho GTPases classification in mammals. Rho GTPases are divided into eight subfamilies, namely the CDC42, RAC, RHO, RHOF, RHOBTB, RHOH, RHOU/RHOV, and RND subfamilies. These are further divided between classical and atypical Rho GTPases. Classical Rho GTPases cycle between an inactive GDP and an activate GTP-bound state while atypical Rho GTPases are mainly regulated through other mechanisms.

**Figure 2 cells-09-01430-f002:**
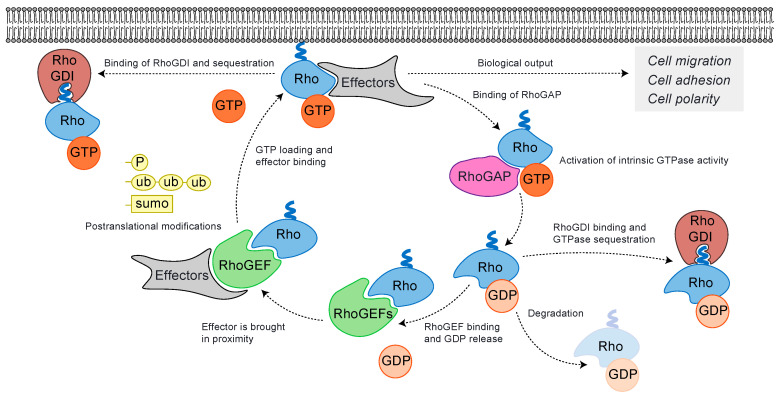
The classical Rho GTPase cycle and its spatiotemporal regulation. RhoGEFs activate Rho GTPases by destabilizing Rho GTPase nucleotide and Mg^2+^ interaction and by promoting GTP loading, which is in excess in cells when compared to GDP. RhoGEFs also act as protein scaffolds by bringing appropriate effectors in close proximity to active Rho GTPases. Once active, Rho GTPases bind with their effectors and trigger biological effects in cells such as cell migration, cell adhesion, and cell polarity. RhoGAPs inactivate Rho GTPases by enhancing their weak intrinsic GTPase activity, which triggers the hydrolysis of GTP in GDP. RhoGDIs bind to Rho GTPases and promote the extraction of the Rho GTPases C-terminal lipid extension from the membrane which becomes ultimately hidden into the RhoGDIs hydrophobic pocket. RhoGDIs are known to sequester Rho GTPases in the cytosol and to prevent their interaction with other regulators. Rho GTPases are also heavily regulated through post-translational modifications, gene expression and their degradation.

**Figure 3 cells-09-01430-f003:**
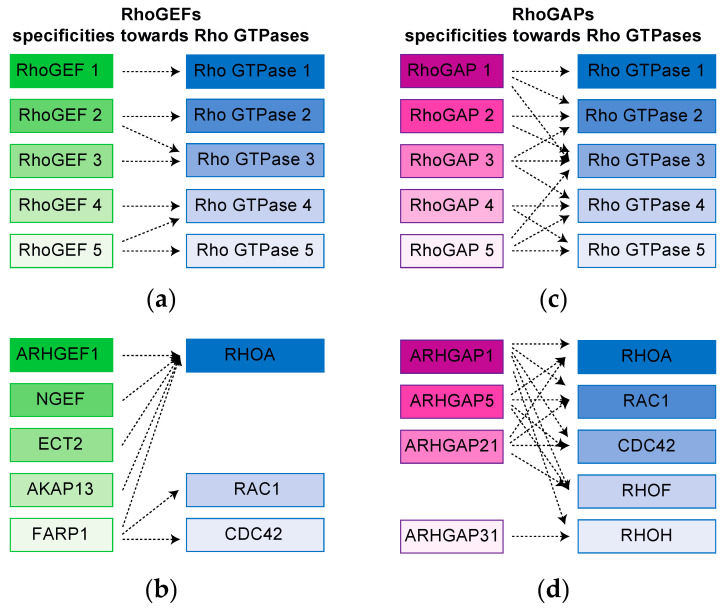
Mapping of Rho GTPases regulators specificities. (**a**) RhoGEFs exhibit specificity towards few Rho GTPases. (**b**) Examples of RhoGEFs and their specificity towards Rho GTPases as mapped by proximity labeling coupled with mass spectrometry. (**c**) RhoGAPs are promiscuous and display broad activity towards Rho GTPases. (**d**) Examples of Rho GAPs and their specificity towards Rho GTPases as mapped by proximity labeling coupled with mass spectrometry.

**Figure 4 cells-09-01430-f004:**
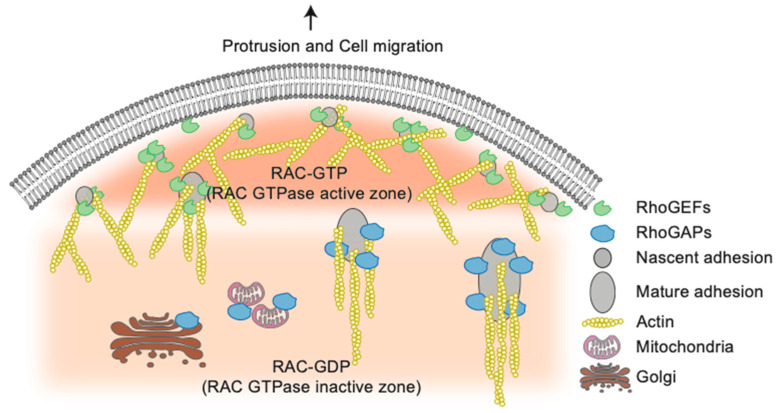
The spatial distribution of Rho GTPases by their regulators the RhoGEFs and RhoGAPs determines Rho GTPase activity. Schematic representation showing that the spatial distribution of RhoGEFs and RhoGAPs helps to create Rho GTPases active and inactive zones in cells The Golgi and mitochondria only host RhoGAPs. This could allow the inactivation of Rho GTPases that would have diffused away from their desired site of action.

**Table 1 cells-09-01430-t001:** Overview of high throughput studies.

Approach	Goal	Cell Types	Reference
qGAP	Identify Rho GTPases binding partners	HEK293 Mouse brain lysate	[61]
BioID	Define Rho family proximity interactome	HEK293 HeLa	[64]
Flag-IP	Define RhoGEFs and RhoGAPs interactome	HEK293	[73]
In vitro siRNA screen	Identify RhoGEFs required for amoeboid movement in melanoma cells	A375M2	[80]
In vitro siRNA screen	Identify RhoGEFs that regulate amoeboid and mesenchymal features	A375M2	[81]
In vitro siRNA screen	Investigate the role of Rho GTPases network components during prostate cancer cell migration	PC3	[21]
In vitro siRNA screen	Identify RhoGEFs and RhoGAPs that regulate breast cancer cell morphology	LM2 MDA-MB-231	[86]
In vitro siRNA screen	Identify RhoGAPs that contribute to EMT	MCF10A	[87]
In vitro siRNA screen	Identify regulators of the endothelial barrier among Rho GTPases network components.	HUVEC	[88]
In vitro shRNA screen	Identify RhoGEFs regulating collective migration	16HBE14o	[89]
In vivo shRNA morphogenesis screen	Identify regulators of skin morphogenesis among Rho GTPases network components	Mouse embryos Primary mouse keratinocytes	[97]

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
