# Peer review of "High Throughput strategies Aimed at Closing the GAP in Our Knowledge of Rho GTPase Signaling"

_cells, 2020, doi:10.3390/cells9061430_

Round 1
Reviewer 1 Report
In this review by Dahmene et al, they present a comprehensive analysis covering various recent major studies using high throughput strategies to unravel Rho GTPase regulatory network signaling. They discuss the most recent literature and current knowledge on the various in vivo and in vitro screening and proteomic approaches used to study Rho GTPase signaling network. Overall, this is a very well-written and organized review that will be of great interest to the scientific community of molecular cell biologists, and in particular to the small GTPase field of research. I have no major issues and it can be accepted as it is.
Reviewer 2 Report
In this review manuscript, the authors revise the recent literature and findings regarding Rho GTPase signalling specificity, with an emphasis in global approaches. This is a very timely review and a complex and important topic. The manuscript covers a lot and still leaves important open questions. I have some minor comments to the manuscript that could potentially improve it and one concern with Figure 4:
- Authors leave out of the Rho GTPase family the Miro proteins. This is ok, but still controversial in the field.
- The introduction and first section of the manuscript are very informative and a good introduction to the field of Rho GTPase signalling. Still, some generalizations about the families leave behind important exceptions, like for example the particularities of RhoB within the Rho subfamily, which, together with other, is modified by farnesylation and regulated by expression, not only GTPase cycle. I understand that the manuscript can not cover all particularities, but some generalizations could be put in context.
- Figure 4 is a paste from another source that includes part of the legend. The source (if other than this original manuscript) needs to be cited and the legend removed. The choice of colours in the text makes it difficult to read.
- The discussion section presents several interesting questions. It comments on the existence of mutations in Rho GTPases in cancer (which are concentrated in a few members) but ignores that there are also mutations identified (and more frequent) in regulators.
Reviewer 3 Report
This is a nicely written review on recent developments around the screening based approaches for characterizing the Rho family GTPases and the regulator interactions. This is a timely review and a useful one for the field especially as we make the transition from single-cell/node-based analyses to large scale systems based methods. There are several points that need to be further addressed before publication of this work.
- Overall, some editorial issues were noted. Please spend some time reviewing carefully typographical and grammatical issues throughout the paper.
- Line 87: “by dislodging GDP…” I was under the impression that GEFs actually dislodge Mg+2 ion instead to allow the nucleotide binding pocket of a GTPase to exchange the guanine nucleotide based on the concentration gradient (cytoplasmic [GTP] >> [GDP]). Based on some reviews by Kent Rossman at UNC, I was under the impression that GEFs were basically Mg+2 pushers…
- Line 97: I would like to see clarifications of the difference between the lipid modification at the caax box versus the charge differences in the hypervariable region and how both may dictate where these GTPases go.
- Line 99/100: “by binding to Rho GTPases’ C-terminal lipid extension…” I thought this was actually a 2-step process in which the GDI binds the Switch I/II of GTPase first, followed by the energetically disfavored step of lipid tail transition from the membrane into the hydrophobic pocket formed by the two beta-globular domains of the GDI… That’s also why if the binding affinity between GTPase-GDI is perturbed a little bit, you get GDI bound to GTPase on the membrane but unable to complete the extraction because it cannot sufficiently overcome this energy barrier… Works by Celine der Mardirossian and Gary Bokoch out of Scripps Institute are nice works in this regard…
- Line 104: I thought Gary Bokoch showed a long time ago that GDI can extract active Rac1 in complex with PAK1 and alluded already to this reallocation of the active GTPase by GDI…
- Line 141: “bypass this problem”: This refers to the issue raised in the previous paragraph. Usually, paragraphs are used to separate ideas; at least the readers expect this. So stylistically speaking, it is not immediately clear when the paragraph does not backward-reference what it is to which it is referring as the problem.
- Line 162: I think “broader specificity” of GAP does not equal rapidity of inactivation. Reaction rate and the substrate specificity are two different things. This statement mixes the two issues.
- Line 233: I suggest to include how the issues raised here could point to why the MMP protease inhibitor treatment may not be therapeutically viable in targeting cancer metastasis. Perhaps modes of motility and invasion driven by different RhoGTPases may be causing this problem.
- Some discussions of caveats in some of these screens would likely be beneficial. Si/shRNA screens can suffer from off-target effects and variable knockdown efficiencies. Is there also note on proximity ligation screens where non-specific interactions and non-direct interactions can be detected?
- Figure 3: Not specific enough to be of much use and can be completely omitted. Or perhaps, show some proven interactions as examples of Rho/GEF/GAP interactions instead of generic items like it is now?
- I found the genetic screening part to be not well integrated at all. It just popped out of nowhere. It is also not mentioned in the abstract at all, leading to the sense of it being out of place. Try to integrate this component better.
- RhoT1&2 are not included. Inclusion of RhoT1 and 2 would make it 22 Rho GTPases…
